# Multimodal Contextual Interactions of Entities: A Modality Circular Fusion Approach for Link Prediction

## ABSTRACT

Link prediction aims to infer missing valid triplets to complete knowledge graphs, with recent inclusion of multimodal information to enrich entity representations. Existing methods project multimodal information into a unified embedding space or learn modality-specific features separately for later integration. However, performance was limited in such studies due to neglecting the modalities compatibility and conflict semantic carried by entities in valid and invalid triplets. In this paper, we aim at modeling inter-entity modality interactions and thus propose a novel **Mo**dality **Ci**rcular fusion approach (**MoCi**), which interweaves multimodal contextual of entities. Firstly, unlike most methods in this task that directly fuse modalities, we design a triplets-prompt modality contrastive pre-training to align modality semantics beforehand. Moreover, we propose a modality circular fusion model using a simple yet efficient multilinear transformation strategy. This allows explicit inter-entity modality interactions, distinguishing it from methods confined to fuse within individual entities. To the best of our knowledge, MoCi presents one of the pioneering frameworks that tailored to grasp inter-entity modality semantics for better link prediction. Extensive experiments on seven datasets demonstrate our model yields SOTA performance, confirming the efficacy of MoCi in modeling inter-entity modality interactions. Our code is released at https://github.com/MoCiGitHub/MoCi.

## CCS CONCEPTS

• **Computing methodologies → Knowledge representation and reasoning**.

## KEYWORDS

Multi-modal Knowledge Graph, Link Prediction, Knowlegde Embedding, Multimodal Fusion

## 1 INTRODUCTION

Link prediction focuses on inferring missing valid triplets [9, 12, 19, 22, 29], represented as <head entity, relation, tail entity>, to enhance the completeness of knowledge graphs (KG). However, traditional link prediction methods face the challenge of structural bias among triplets, which impacts the inferencing performance. To mitigate these issues, multimodal contexts[8, 15, 24], including text and images, have been integrated to enhance entity representations.

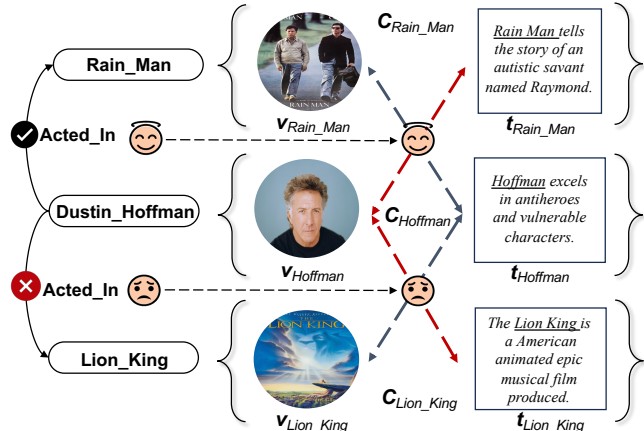

**Figure 1: Examples of multimodal contextual interactions of entities. The triplet (*Dustin_Hoffman, acted_in, Rain_Man*) is valid, while (*Dustin_Hoffman, acted_in, The_Lion_King*) is invalid. $C_{Hoffman}$ indicates the multimodal context.**

For example, a textual description $t_1$ and visual image $v_1$ form the multimodal context of entity $e_1$, denoted as $C_{e_i} = \{t_i, v_i\}$. Multimodal contexts capture rich information about entities and are leveraged to improve link prediction[29].

In this context, several models have been proposed to utilize multimodal information for link prediction. Among these, some studies have focused on projecting modality-specific information into a unified embedding space and incorporating it within entity embeddings [18, 26, 37], as depicted in entity-level modality aggregation models (Figure 2). However, this approach might lead to the loss of unique information specific to each modality. In response, as depicted in Figure 2, modality split and ensemble models have been proposed to address this limitation. These models emphasize the independent learning of distinct modality-specific features followed by their integration during the inference stage [34, 39]. Nonetheless, the modality interactions in these methods are typically confined within individual entity (entity-specific modality fusion). Thus neglecting the rich semantic information that can be derived from interactions between multimodal contexts of different entities, which could potentially contribute to the evaluation of triplets.

Figure 1 provides illustrations of multimodal contextual interactions of entities. Take the scenario in Figure 1 as an example, where the triplet $tp_1$ (*Dustin_Hoffman, acted_in, Rain_Man*) is a valid triplet, while $tp_2$ (*Dustin_Hoffman, acted_in, The_Lion_King*) is an invalid triplet. The entities involved carry multimodal contexts, including textual and visual information, represented as $C_{hoffman} = \{t_{hoffman}, v_{hoffman}\}$. In contexts where $tp_1$ is valid and $tp_2$ is not, the textual content $t_{hoffman}$ ("Hoffman excels in

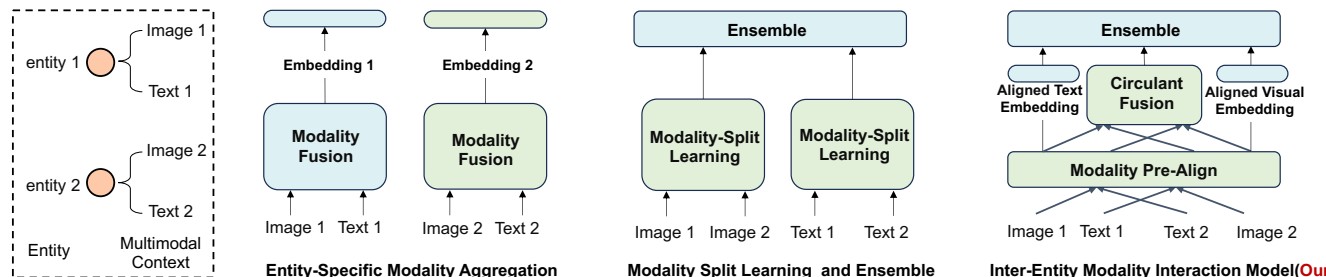

**Figure 2: Structural comparisons among different approaches for Link Prediction. Right: Entity-Specific Modality Aggregation. Middle: Modality Split Learning and Ensemble. Left: Inter-Entity Modality Interaction Model(Ours).**

antiheroes and vulnerable characters") exhibits semantic incompatibility with the visual information $v_{The\_Lion\_King}$ to a greater extent than with $v_{Rain\_Man}$. Conversely, the semantic compatibility between $t_{hoffman}$ and $v_{Rain\_Man}$ surpasses that between $t_{hoffman}$ and $v_{The\_Lion\_King}$. A similar pattern of semantic compatibility and incompatibility is observable in the visual context $v_{hoffman}$ with respect to the textual descriptions of $t_{Rain\_Man}$ and $t_{The\_Lion\_King}$, which are carried by the validity or invalidity of triplets. This implies that the semantic among different modalities of entities is consistent with the validity or invalidity of the triplets formed by these entities.

Inspired by this, we propose to extract and learn semantic critical from the multimodal contexts of different entities (inter-entity modality interaction) for link prediction, rather than focusing solely on intra-entity modality interactions. To this end, we introduce a **Mo**dality **Ci**rcular contrastive and fusion approach (**MoCi**). Specifically, a triplets-prompt contrastive pre-training strategy is introduced to align semantics across modalities. Subsequently, we employ a multilinear transformation strategy for modality circular fusion inter entity. Finally, relational context-aware predictions are integrated across modalities. To the best of our knowledge, MoCi represents a pioneering framework that shifts modality fusion in link prediction to the inter-entity perspective, thus significantly advancing accuracy in link prediction. Our contributions are as follows:

- MoCi is the first work to advance multimodal fusion in the link prediction task toward an inter-entity perspective.
- Proposed triplets-prompt modality contrastive pre-training aligns modality semantics in advance, rather than direct modality fusion in most existing methods, addressing potential semantic gaps between modalities.
- We propose an efficient modality circular fusion model using multilinear transformation strategy. This allows explicit inter-entity modality interactions, setting it apart from methods confined to fuse within individual entities.
- Our approach was validated across seven datasets, marking the most extensive experimentation to date. Experimental results demonstrate that MoCi achieves state-of-the-art performance, confirming the effectiveness of proposed method in modeling inter-entity modality interactions.

## 2 RELATED WORK

### 2.1 Uni-modal Link Prediction

Uni-modal link prediction utilizes the structural topology of KG and embeddings to predict missing links. Models like RotatE[47] interpret relations through complex space rotations, effectively handling various relational properties, while ANALOGY[11] maintains analogical consistency to boost reasoning. Additional models include TransE[3], DistMult[4], ComplEx[31], GC-OTE[40], TuckER[14], ConvE[30], PairRE[17], MDCN[44], WGCN[7], and JointE[45], each offering distinct approaches to link prediction. The multimodal information of entities is not taken into account in these methods.

### 2.2 Entity-level Modality Aggregation

Entity-level modality aggregation models improve link prediction by incorporating multimodal information such as visual images and text description to enrich entity representations. For example, HRGAT[27] uses a hierarchical graph attention network to integrate modalities effectively, while IKRL[25] combines images with structured data to improve visual entity recognition. Additional models like MKGformer[32], OTKGE[49], MKBE[22], QEB[33], CMGNN[23], TBKGC[13], RSME[20], TransAE[48], VBKGC[42], MULTIFORM[36] and AdaMF-MAT[43] also utilize diverse modalities but often struggle to preserve the unique properties of each modality when merging them into a single embedding space.

### 2.3 Modality Split and Ensemble

Modality split and ensemble models focus on learning modality specific representations and integrating predictions from different modalities in knowledge graphs. MoSE[39] pioneered this framework by introducing an ensemble method that dynamically assesses and integrates the importance of different modalities during inference. Building on this foundation, the IMF[34] model enhances the approach by using a bilinear pooling module. Nonetheless, such models usually restrict the interactions to within individual entity. The performance was limited in such studies due to neglecting the modalities compatibility and conflict semantics carried by entities in valid and invalid triplets. The introduction of MoCi aims to address this deficiency. It presents the first framework tailored to capture inter-entity modality semantics for significantly improving link prediction.

**Table 1: Notations and Definitions**

| Notations | Descriptions |
|---|---|
| MoCi | Our model name |
| $E, R, T$ | Set of entities, relations, and triplets |
| $M = \{s, v, t\}$ | Set of modalities |
| $|E|, |R|, |M|$ | Length of set |
| $e, r$ | Entity and relation |
| $\mathbf{e}^m$ | Features vectors of entity $e$ in modality $m$ |
| $\mathbf{E}^m$ | Feature matrix in modality $m$ |
| $\mathcal{X}^{|E| \times |D|}_{|M|}$ | Tensor $\mathcal{X}$ with size $|M| \times |E| \times |D|$ |

## 3 DEFINITIONS

A knowledge graph is formalized as $G = \{E, R, T\}$, where $E, R$ denotes entity set and relation set, and $T = \{(h, r, t)|h, t \in E, r \in R\}$ refers to triplets that describe relations between entities. In multimodal KG, each entity is associated with multi-modal context, including textual, visual, and structural information. We define the set of modalities as $M = \{s, v, t\}$, where $s,v,t$ denote structural, visual, and textual, respectively. The objective of link prediction is to infer missing triplets according to the existing factual triplets $T$ and multi-modal contexts, which could be generally represented as $(?, r, t)$ or $(h, r, ?)$, namely predicting the possible subject or object. Notations and their definitions are given in Table 1.

## 4 METHODOLOGY

In this section, we offer a comprehensive introduction to our proposed models, which is designed to capture inter-entity modality semantics to enhance link prediction. This includes modality-specific embeddings initialization, triplets-prompt modality contrastive pre-train, modality circular fusion, and relational context-aware prediction. The overall framework of the proposed model is illustrated in Figure 3.

### 4.1 Modality-Specific Embeddings Initialization

Distinguishing from traditional neural networks that directly take raw data features as input and process them through large-scale nonlinear networks, we introduce a novel strategy here: the features of each modality are defined as learnable parameters, and the corresponding parameter embedding is initialized by the entity modality features. In this way, we effectively avoid the model's excessive dependence on deep nonlinear networks while graph is highly sparse. Additionally, this initialization method better retains and utilizes the original feature information of each entity, which encourage the model converge to the appropriate solution faster. In this subsection, we explore the initialization strategies for the textual, and visual, structural modalities as follows:

**Textual Modality:** Entity text descriptions usually contain some valuable textual information, typically including name, type, attribute etc. We utilize the BERT-Base[5] model pre-trained on a large-scale corpus to capture word contextual semantic relations and generate high-quality text feature representations on diverse datasets, such as YAGO15K[29], VTKG-I[16] and MKG-Y[9]. This approach provides more accurate textual semantic descriptions for subsequent tasks.

**Visual Modality:** For the visual description of each entity, the pretrained VGG16[28] model is utilized for feature extraction on YAGO15K and DB15K datasets. Considering the diverse complexity of the visual content of different datasets, such as VTKG-I, VTKG-V, and WN18RR++, the Vision Transformer (ViT[2]) model is adopted to generate more expressive feature representations, which could effectively capture dependencies between different regions of given complex images through their self-attention mechanism.

**Structural Modality:** KG is essentially a topological structure that reflects the connection relation between known entities. Different from the visual and textual attributes of the entity itself, the structural features could further reflect the association characteristics between different entities. Hence, we employ a multi-relational graph convolutional network (MR-GCN[46]) to learn the graph structure and obtain the initial node embeddings.

The embeddings for textual, visual and structural modalities are initialized with features extracted from models ideally suited to their respective characteristics. This detailed strategy is further discussed in Section 5.1, outlines our experimental setup and the specific models employed for link prediction tasks.

### 4.2 Triplets-Prompt Modality Contrastive Pre-train

Most existing methods typically directly fuse different modalities [9, 12, 22, 27, 34, 39] without accounting for potential semantic differences between them. We propose a pre-training strategy before modal feature fusion. The purpose of this strategy is to align modality semantics in advance and interact modality information between entities.

**Triplets-Prompt Modality Contrastive Learning:** In this section, we introduce a novel cross-modal contrastive learning in the context of triplets, which utilizes the semantic compatibility/incompatibility carried by valid/invalid triplets. Concretely, this strategy focuses on triplet-prompt modal contrastive learning, and carefully selects high-confidence positive samples from the existing factual (valid) triplets and negative samples from the invalid context. This strategy also allows for efficient learning of graph structure by leveraging triplets without the need for costly inclusion of all nodes.

Formally, the interaction patterns of entities in valid triplets $(h, r, t)$ and invalid triplets $(h, r, t')$, i.e. $sim$ and $dsim$ are modeled as follows:

$$sim(\mathbf{a}^p_h, \mathbf{b}^q_t) = cos(\mathbf{a}^p_h, \mathbf{b}^q_t)^2 \tag{1}$$

$$dsim(\mathbf{a}^p_h, \mathbf{b}^q_{t'}) = \max(0, d - cos(\mathbf{a}^p_h, \mathbf{b}^q_{t'}))^2. \tag{2}$$

where $p, q \in M$ are different modalities within the modalities set $M$, which includes structural ($s$), textual ($t$), and visual ($v$) modalities. Additionally, $\mathbf{a}^p_h = concat(\mathbf{e}^p_h, \mathbf{r}^p_r) \in \mathbb{R}^{2d}$ is constructed by concatenating head entity features $\mathbf{e}^p_h \in \mathbb{R}^d$ with corresponding relational features $\mathbf{r}^p_r \in \mathbb{R}^d$, and $\mathbf{b}^q_t = AggRC(\mathbf{e}^q_t) \in \mathbb{R}^{2d}$ represents the tail entities' representations after being enriched by the aggregate relation context ($AggRC$). More details could be referred to in Section 4.4 for an in-depth discussion on $\mathbf{b}^q_t$. And the margin $d$ distinguishes between different interaction pattern metrics used

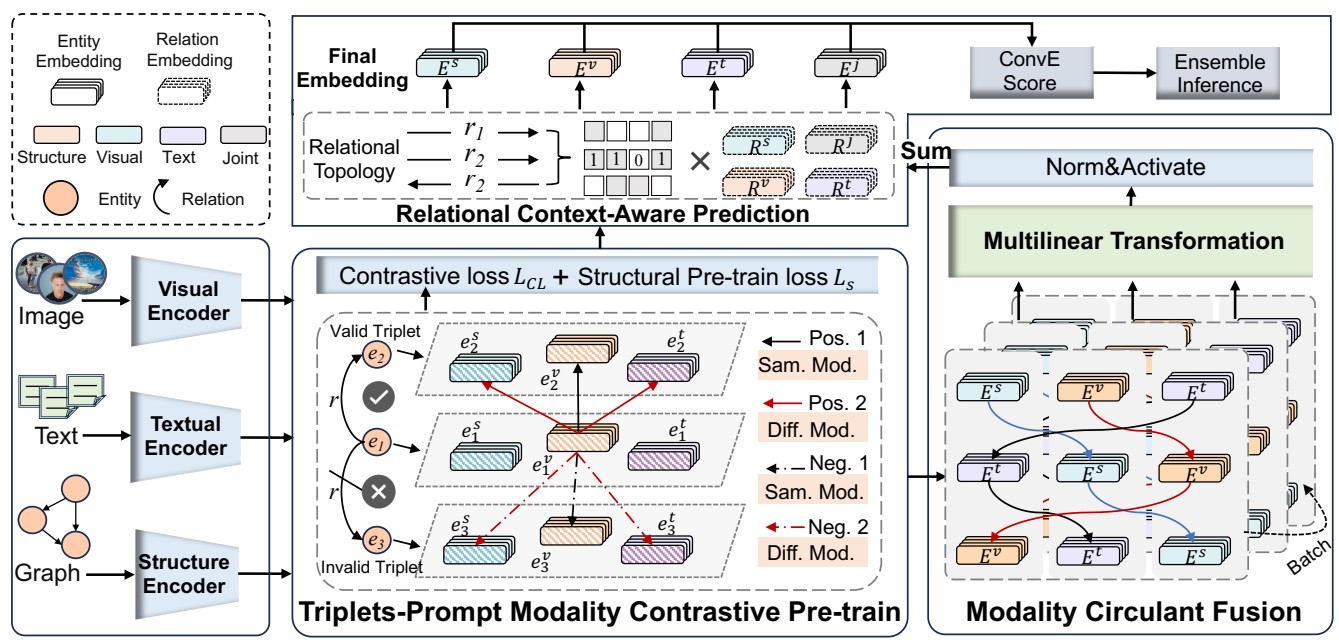

**Figure 3: The framework of our method, MoCi, considers multimodal information $(s, v, t)$. Initial features are derived from modality-specific embeddings initialization (Sec. 4.1), followed by contrastive pre-training (Sec. 4.2). Pre-aligned semantic features are processed through the modality circular fusion module (Sec. 4.3). Then, these modality features and joint features are input into the relational context-aware prediction module (Sec. 4.4).**

to align inter-entity modality semantics, refining our model's contrastive learning dynamic tradeoff. Cosine similarity ($cos$) is defined as follows:

$$cos(\mathbf{a}_h^p, \mathbf{b}_t^q) = \frac{\mathbf{a}_h^p \cdot \mathbf{b}_t^q}{|\mathbf{a}_h^p||\mathbf{b}_t^q|}, \tag{3}$$

Building on the above, we introduce the overall triplet contrastive loss $\mathcal{L}_{cL}$ as follows:

$$\mathcal{L}_{cL} = \frac{1}{N \times |O|} \sum_{(p,q) \in O} \sum_{i=1}^{N} \left[ sim(\mathbf{a}_{h_i}^p, \mathbf{b}_{t_i}^q) + \sum_{k \neq i} dsim(\mathbf{a}_{h_i}^p, \mathbf{b}_{t_k}^q) \right] \tag{4}$$

where $O = \{(p, q) \mid p, q \in M\}$ represents all modality combinations, and $N$ is the total number of triplets.

**Pre-training Loss:** In $\mathcal{L}_{cL}$, we have ensured inter-entity modality interaction. Furthermore, to enhance structural learning and achieve clearer structural embeddings, we have incorporated contrastive and structural losses to define our pre-training loss $\mathcal{L}_{pre}$ as follows:

$$\mathcal{L}_{pre} = \mathcal{L}_{cL} + \mathcal{L}_s \tag{5}$$

where $\mathcal{L}_s = f^s(h, r, t)$ is the structural loss function, with implementation details provided in Section 4.4.

## 4.3 Modality Circular Fusion

The highly sparsity of KG triplets directly affects the design of subsequent network architecture, such as $10e^{-6}$ on YAGO15K and DB15K datasets[29], etc., which is calculated by dividing the total

number of known triplets by that of possible triplets. Hence, how to effectively further avoid deepening the nonlinear inference layers under good parameter initialization conditions, becomes the focus of this section. To this end, we define a set of modality-shared linear transformation parameters, and achieve efficient interaction of multi-modal contexts through a circulant fusion strategy based on Tensor product [41, 46]. This allows explicit inter-entity modality interactions, distinguishing it from methods confined to fuse within individual entities.

Specifically, we employ a simple yet effective multilinear transformation strategy inspired by the Tensor product, in which each element denotes a tube rather than a scalar in the conventional matrix. The Tensor product of tensor $\mathcal{X} \in \mathbb{R}_M^{N \times D}$ and corresponding linear transformation parameter tensor $\mathcal{W} \in \mathbb{R}_M^{D \times D'}$ is defined as:

$$\mathcal{Z} = \mathcal{X} * \mathcal{Y} = fold(bcirc(\mathcal{X}) \cdot unfold(\mathcal{W})) \tag{6}$$

where $\mathcal{Z} \in \mathbb{R}_M^{N \times D'}$, $*$ indicates the Tensor product operator. The $unfold(.)$ operator flattens tensor $\mathcal{X} \in \mathbb{R}_M^{N \times D}$ to a matrix of size $MN \times D$, and $fold(\cdot)$ corresponds to its inverse. $bcirc(\mathcal{X}) \in \mathbb{R}^{MN \times MD}$ is the unfolding block circulant matrix of $\mathcal{X}$. For more detailed explanations about the Tensor product, please refer to Reference [41, 46]. In such a paradigm, we formulate the modality circular fusion as:

$$\mathcal{F} = \mathcal{E} * \mathcal{W} \tag{7}$$

where $\mathcal{E} \in \mathbb{R}_{|M|}^{|E| \times |D|}$ is stacked from $|M|$ modalities, $|D|$ denotes dimension of features. Each frontal slice $\mathbf{E}^{(\mathbf{m})} \in \mathbb{R}^{|E| \times |D|}$ (i.e. features of modalities $m$) in $\mathcal{E}$ is transformed linearly by learnable

parameters $\mathbf{W}^{(m)} \in \mathbb{R}^{|D| \times |D'|}$ of $\mathcal{W}$, as general formulated in Equation (8):

$$\mathbf{F}^{(m)} = \mathbf{E}^{(m)} \cdot \mathbf{W}^{(1)} + \mathbf{E}^{(m-1)} \cdot \mathbf{W}^{(|M|-m+2)} + \ldots + \mathbf{E}^{(m+1)} \cdot \mathbf{W}^{(2)} \quad (8)$$

This implies that there are $|M|$ different combination strategies between $\mathbf{E}^{(i)} \in \mathbb{R}^{|E| \times |D|}$ and $\mathbf{W}^{(j)} \in \mathbb{R}^{|D| \times |D'|}$, determined by the way the block circulant matrix $bcirc(\mathcal{E})$ generated. Hence, the derived embedding tensor $\mathcal{F}$ follows a circulant formulation to fuse features from different modalities roundly, and also ensure semantic consistency encoding among different modalities by parameter sharing. Building on the above, the final joint features $\mathbf{J}$ are obtained as follows:

$$\mathbf{J}_{ik} = \frac{e^{\sum_{m=1}^{|M|} \mathcal{F}_{i,k,m}}}{\sum_{l=1}^{|D'|} e^{\sum_{m=1}^{|M|} \mathcal{F}_{i,l,m}}} \quad (9)$$

The joint features $\mathbf{J}$ then undergo normalization and are processed through activation functions.

**Scalability**: Our model MoCi includes three commonly used modalities of entities, i.e. $M = \{s, v, t\}$. It is straightforward to generalize the proposed method for fusing more modalities by extending $M$. Moreover, despite the parameter $\mathcal{W}$ being a tensor, its size is merely *number of modalities* × *feature dimension* × *feature dimension(transformed)*, where the number of modalities $|M| = 3$ in our work. This ensures efficiency and scalability even including more modalities.

**Explainability**: The proposed circular fusion operation goes beyond merely aggregating matrix multiplication results from corresponding frontal slices in the spatial domain. It embodies the essence in Fourier domain since $\mathcal{E} * \mathcal{W} = IFFT(FFT(\mathcal{E}) \Delta FFT(\mathcal{W}))$, where $\Delta$ indicates slice-by-slice matrix multiplication. This implies the linear fusion across modalities in the Fourier domain, thereby offering a nuanced capture of inter-modality correlations and complementarity. For more detailed explanations about the Tensor product, please refer to Reference [41, 46].

## 4.4 Relational Context-Aware Prediction

Relations as the most essential description of the association between entities. Therefore, our proposed model further considers relations in the prediction phase. We enrich entity representations through aggregate relation context by extend [21].

**Aggregate Relation Context (AggRC):** In our model, the contextual relation feature $\mathbf{c} \in \mathbb{R}^{2|R|}$ is employed to accurately capture the relational topology of the entity-oriented graph. The corresponding feature is defined as follows:

$$\mathbf{c}_{e,r} = \begin{cases} H(e,r), & \text{if } r \leq |R|, \\ T(e,r), & \text{if } |R| < r \leq 2|R|, \end{cases} \quad (10)$$

where $|H(e,r)| = \{(e,r,e_k) \mid \exists e_k, (e,r,e_k) \in T\}$ denotes the frequency of entity $e$ serving as the head of the relation $r$, and $|T(e,r)| = \{(e_k,r,e) \mid \exists e_k, (e_k,r,e) \in T\}$ represents the frequency of $e$ serving as the tail of relation $r$. This bidirectional representation ensures that the connections between each entity and all relations within the graph are comprehensively captured, thereby providing a rich relational context for the model.

Entity representation in each modality is further enhanced through the aggregation of relation context information for entities as:

$$AggRC(\mathbf{e}^{m'}) = concat(\mathbf{e}^{m'}, \mathbf{c}^{\top} \mathbf{R}^{m'})) \quad (11)$$

where $m' \in \{s, v, t, j\}$ corresponds to structural ($s$), visual ($v$), textual ($t$) modalities and joint modalities ($j$) that obtained from the implement of modality circular fusion. Relation embedding $\mathbf{R} \in \mathbb{R}^{2|R| \times d}$ encompasses both forward and reverse relations. The forward relation can be represented as $(h, r^+, t)$, while the reverse relation can be represented as $(t, r^-, h)$.

**Training Loss:** We extendt the score function of the classical ConvE[30] model by aggregating relation context into entity representations. Given a triplet $(h, r, t)$, the prediction score is defined as:

$$f^{m'}(h,r,t) = \sigma(vec(\sigma([vec(\mathbf{b}_h^{m'} \mathbf{W}_h^{m'}); vec(\mathbf{r}_r^{m'})] * \omega)) \mathbf{W}^f) \mathbf{e}_t^{m'} \quad (12)$$

where $\mathbf{b}^{m'} = AggRC(\mathbf{e}^{m'}) \in \mathbb{R}^{2d}$, represents the entities' representations after being enriched by the aggregate relation context $(AggRC)$. $\mathbf{W}_h^{m'} \in \mathbb{R}^{2d \times d}$ represents the transformation matrix. The convolutional filter is represented by $\omega$, $\mathbf{W}^f$ the is weight of the fully connected layer. $\sigma$ denotes a non-linear activation function such as sigmoid or ReLU, which endows the model with the capacity to discern complex, non-linear relations present within the dataset.

To optimize the model, a total loss function is defined to include losses from multiple modalities and the joint modalities ($j$), as shown below:

$$\mathcal{L} = \mathcal{L}_s + \mathcal{L}_v + \mathcal{L}_t + \mathcal{L}_j \quad (13)$$

Each loss term $\mathcal{L}_*$ corresponds to a specific modality, aiming to accurately grasp inter-entity modality semantics for better link prediction.

**Table 2: Statistical Information of Seven Multimodal Benchmark Datasets**

| Datasets | $|E|$ | $|R|$ | Train | Valid | Test |
|---|---|---|---|---|---|
| YAGO15K | 15,283 | 32 | 86,020 | 12,289 | 24,577 |
| DB15K | 14,777 | 279 | 69,319 | 9,903 | 19,806 |
| VTKG-I | 181 | 217 | 1054 | 131 | 131 |
| VTKG-C | 43,267 | 2,731 | 89190 | 11149 | 11152 |
| WN18RR++ | 41,105 | 11 | 86835 | 3034 | 3134 |
| MKG-W | 15000 | 169 | 34196 | 4276 | 4274 |
| MKG-Y | 15000 | 28 | 21310 | 2665 | 2663 |

**Table 3: Feature Extraction Models and Multimodal Characteristics of Datasets**

| Datasets | $T - PM$ | $V - PM$ | $E^T$ | $E^V$ | $R^T$ | $R^V$ |
|---|---|---|---|---|---|---|
| YAGO15K | BERT-Base | VGG16 | ✓ | ✓ | ✗ | ✗ |
| DB15K | BERT-Base | VGG16 | ✓ | ✓ | ✗ | ✗ |
| VTKG-I | BERT-Base | ViT | ✓ | ✓ | ✓ | ✓ |
| VTKG-C | BERT-Base | ViT | ✓ | ✓ | ✓ | ✓ |
| WN18RR++ | BERT-Base | ViT | ✓ | ✓ | ✓ | ✗ |
| MKG-W | SBERT | BEiT | ✓ | ✓ | ✗ | ✗ |
| MKG-Y | SBERT | BEiT | ✓ | ✓ | ✗ | ✗ |

**Table 4: Comparative Analysis of Knowledge Graph Completion on DB15K, YAGO15K, MKG-Y and MKG-W.**

| Model | YAGO15K | | | | DB15K | | | |
|---|---|---|---|---|---|---|---|---|
| | MRR | H@1 | Hit@3 | H@10 | MRR | H@1 | Hit@3 | H@10 |
| TransE | 0.1610 | 0.0510 | - | 0.3840 | 0.2560 | 0.1370 | - | 0.4690 |
| ConvE | 0.2670 | 0.1680 | - | 0.4260 | 0.3120 | 0.2190 | - | 0.5070 |
| TuckER | 0.2810 | 0.1830 | - | 0.4570 | 0.3410 | 0.2430 | - | 0.5380 |
| IKRL | 0.1390 | 0.0480 | - | 0.3170 | 0.2220 | 0.1110 | - | 0.4260 |
| MKGC | 0.1290 | 0.0410 | - | 0.2970 | 0.2080 | 0.1080 | - | 0.4190 |
| MKBE | 0.2730 | 0.1750 | - | 0.4230 | 0.3320 | 0.2350 | - | 0.5130 |
| IMF | 0.3120 | 0.2339 | 0.3432 | 0.4617 | 0.4410 | 0.3784 | 0.4721 | 0.5618 |
| MANS | - | - | - | - | 0.3320 | 0.2040 | 0.4200 | 0.5500 |
| AdaMF-MAT | - | - | - | - | 0.3514 | 0.2530 | 0.4111 | 0.5292 |
| our (w/o MCF) | 0.3594 | 0.2931 | 0.3925 | 0.4874 | 0.4227 | 0.3614 | 0.4517 | 0.5384 |
| our (w/o PCL) | 0.4189 | 0.3560 | 0.4508 | 0.5341 | 0.4578 | 0.3911 | 0.4890 | 0.5865 |
| our (w/o AggRC) | 0.4244 | 0.3602 | 0.4560 | 0.5438 | 0.4612 | 0.3959 | 0.4909 | 0.5871 |
| **our** | **0.4303** | **0.3688** | **0.4617** | **0.5449** | **0.4621** | **0.3977** | **0.4935** | **0.5889** |

| Model | MKG-Y | | | | MKG-W | | | |
|---|---|---|---|---|---|---|---|---|
| | MRR | H@1 | Hit@3 | H@10 | MRR | H@1 | Hit@3 | H@10 |
| GC-OTE | 0.3295 | 0.2677 | 0.3644 | 0.4408 | 0.3392 | 0.2655 | 0.3596 | 0.4605 |
| IKRL | 0.3322 | 0.3037 | 0.3428 | 0.3826 | 0.3236 | 0.2611 | 0.3475 | 0.4407 |
| TBKGC | 0.3399 | 0.3047 | 0.3527 | 0.4007 | 0.3148 | 0.2531 | 0.3398 | 0.4324 |
| MMKRL | 0.3681 | 0.3166 | 0.3979 | 0.4531 | 0.3010 | 0.2216 | 0.3409 | 0.4469 |
| RSME | 0.3444 | 0.3178 | 0.3607 | 0.3909 | 0.2923 | 0.2336 | 0.3197 | 0.4043 |
| OTKGE | 0.3551 | 0.3197 | 0.3718 | 0.4138 | 0.3436 | 0.2885 | 0.3625 | 0.4488 |
| KBGAN | 0.2971 | 0.2281 | 0.3488 | 0.4021 | 0.2947 | 0.2221 | 0.3487 | 0.4064 |
| MANS | 0.2903 | 0.2525 | 0.3135 | 0.3449 | 0.3088 | 0.2489 | 0.3363 | 0.4178 |
| MMRNS | 0.3593 | 0.3053 | 0.3907 | **0.4547** | 0.3413 | 0.2737 | 0.3748 | **0.4682** |
| IMF | 0.3580 | 0.3300 | 0.3710 | 0.4060 | 0.3450 | 0.2880 | 0.3660 | 0.4540 |
| our (w/o MCF) | 0.3546 | 0.3162 | 0.3763 | 0.4204 | 0.3151 | 0.2536 | 0.3416 | 0.4269 |
| our (w/o PCL) | 0.3761 | 0.3481 | 0.3903 | 0.4288 | 0.3342 | 0.2846 | 0.3547 | 0.4314 |
| our (w/o AggRC) | 0.3796 | 0.3494 | 0.3960 | 0.4367 | 0.3440 | 0.2912 | 0.3673 | 0.4438 |
| **our** | **0.3887** | **0.3562** | **0.4041** | 0.4495 | **0.3581** | **0.3074** | **0.3801** | 0.4593 |

## 5 EXPERIMENTS

### 5.1 Datasets

Our model was evaluated on seven diverse multimodal publicly available datasets: YAGO15K, DB15K, VTKG-I, VTKG-C, WN18RR++, MKG-W and MKG-Y. YAGO15K and DB15K datasets are obtained from MMKG[29]; VTKG-I, VTKG-C and WN18RR++ datasets are obtained from VISTA[16]; MKG-W and MKG-Y datasets are obtained from MMRNS[9]. Table 2 delivers a detailed statistical overview. And the feature extraction models used for each dataset, along with the availability of textual and visual features within entities and relations, are shown in Table 3. The acronyms $T-PM$ and $V-PM$ denote the models used for textual and visual feature extraction, respectively. The notations $E^T, E^V, R^T$, and $R^V$ specify the availability of textual, visual features for entities ($E$) and relations ($R$).

### 5.2 Baselines

To substantiate the effectiveness of our model, we selected 23 prominent link prediction models proposed in recent years as our baselines. For clarity in comparison, these models are categorized into unimodal knowledge graph embedding models and multimodal knowledge graph embedding models.

- **Unimodal Models:** This category includes models such as TransE [3], DistMult [4], RotatE [47], PairRE [17], GC-OTE [40], TuckER [14], ConvE [30], ComplEx [31], and ANALOGY [11], which focus exclusively on the structural learning of given KG.
- **Multimodal Models:** This category includes models such as IKRL[25], MKBE[22], TransAE[48], AdaMF-MAT[43], MMKRL [35], KBGAN[6], MMRNS[10], MKGC[12], OTKGE[49], TBKGC [13], RSME[20], MANS[38], MoSE[39], and IMF[34], which focus on integrating multimodal data to enhance KG embeddings.

**Table 5: Comparative Analysis of Knowledge Graph Completion on VTKG-I, VTKG-C and WN18RR++.**

| Model | VTKG-I | | | | VTKG-C | | | | WN18RR++ | | | |
|---|---|---|---|---|---|---|---|---|---|---|---|---|
| | MRR | H@1 | Hit@3 | H@10 | MRR | H@1 | Hit@3 | H@10 | MRR | H@1 | Hit@3 | H@10 |
| ANALOGY | 0.3040 | 0.2328 | 0.3015 | 0.4466 | 0.2963 | 0.2609 | 0.3180 | 0.3532 | 0.4128 | 0.3969 | 0.4175 | 0.4438 |
| ComplEx-N3 | 0.3911 | 0.3168 | 0.4046 | 0.5191 | 0.3944 | 0.3515 | 0.4079 | 0.4815 | 0.4745 | 0.4292 | 0.4895 | 0.5675 |
| RotatE | 0.3131 | 0.2099 | 0.3473 | 0.5267 | 0.3893 | 0.3473 | 0.4062 | 0.4704 | 0.4606 | 0.4274 | 0.4754 | 0.5230 |
| PairRE | 0.4104 | 0.3015 | 0.4504 | 0.6145 | 0.3876 | 0.3431 | 0.4013 | 0.4782 | 0.4529 | 0.4127 | 0.4663 | 0.5351 |
| RSME | 0.4027 | 0.3321 | 0.4122 | 0.5573 | 0.3942 | 0.3513 | 0.4096 | 0.4776 | 0.4567 | 0.4175 | 0.4751 | 0.5300 |
| TransAE | 0.2437 | 0.0687 | 0.3092 | 0.6374 | 0.0751 | 0.0053 | 0.1053 | 0.2072 | 0.0900 | 0.0040 | 0.1291 | 0.2511 |
| OTKGE | 0.4278 | 0.3588 | 0.4466 | 0.5458 | 0.3939 | 0.3446 | 0.4152 | 0.4881 | 0.4327 | 0.3722 | 0.4663 | 0.5407 |
| MoSE-AI | 0.4306 | 0.3473 | 0.4466 | 0.6221 | 0.3929 | 0.3186 | 0.4301 | **0.5210** | 0.4857 | 0.4255 | 0.5094 | 0.5996 |
| IMF | 0.4184 | 0.3282 | 0.4656 | 0.5649 | 0.4116 | 0.3706 | 0.4261 | 0.4935 | 0.4749 | 0.4397 | 0.4845 | 0.5469 |
| our (w/o MCF) | 0.3955 | 0.3206 | 0.4237 | 0.5382 | 0.3883 | 0.3522 | 0.4022 | 0.4586 | 0.4620 | 0.4314 | 0.4750 | 0.5282 |
| our (w/o PCL) | 0.4615 | 0.3740 | 0.4733 | 0.6565 | 0.4148 | 0.3705 | 0.4297 | 0.5028 | 0.5017 | 0.4561 | 0.5155 | 0.5949 |
| our (w/o AggRC) | 0.4709 | 0.3779 | 0.5082 | 0.6641 | 0.4186 | 0.3741 | 0.4335 | 0.5082 | 0.5105 | 0.4647 | 0.5244 | 0.6008 |
| our | **0.4779** | **0.3969** | **0.5153** | **0.6718** | **0.4234** | **0.3770** | **0.4405** | 0.5169 | **0.5149** | **0.4687** | **0.5303** | **0.6112** |

## 5.3 Implementation Details

Our experiments were conducted on an NVIDIA RTX A6000 GPU with 48GB of RAM, utilizing the PyTorch[1] deep learning framework for implementation. Throughout the training process, we configured the number of training epochs to 1,000, with a batch size of 256, modality embedding dimensions set at 256, and a learning rate of 0.0005. For baseline methods, we relied on both their originally reported results and our reproduction of those results. This experimental setup ensures a rigorous and fair comparison across all evaluated models.

## 5.4 Overall Performance Analysis

MoCi has a significant performance improvement compared with all baseline models, which fully demonstrates the effectiveness of the proposed methods. As illustrated in Tables 4 and 5, the MoCi model shows state-of-the-art performance on multiple evaluation metrics. In details, on the YAGO15K dataset, MoCi achieves 11.83% improvement in MRR over the current SoTA approaches, along with improvement of 13.49% in H@1, 11.85% in H@3, and 8.32% in H@10. Furthermore, on the VTKG-I dataset, MoCi achieves 4.73% improvement in MRR compared with the suboptimal approaches, along with enhancements of 4.96% in H@1, 6.87% in H@3, and 4.97% in H@10. Moreover, MoCi also shows significant improvement on the other five datasets.

Although unimodal models show more and more complex architectural designs to further enhance the ability to capture the relations between entities in KG, they still face performance limitations due to potential structural biases. Different from these methods, our MoCi effectively alleviates the structural bias of KG and improves the performance link prediction by effectively exploiting the interaction of multimodal complementary information. Moreover, it is worth noting that our model can still achieve a competitive performance level with the unimodal methods as shown in Table 6,

even when only structural information is utilized, which further confirms the effectiveness of our model on structure learning.

Multimodal models leverage information from diverse modalities, yet the fusion of modality features is often constrained within individual entities, thereby limiting performance. As opposed to such models, our approach further emphasizes learning inter-entity modality interactions in addition to intra-entity modality interactions. This comprehensive interaction approach enables MoCi to effectively model and utilize the commonalities and complementarities between different modalities, leading to significant improvements in link prediction accuracy.

## 5.5 Module Ablation Study

To ascertain the importance of each module in MoCi, we conducted a module ablation study by removing certain components: removing the multimodal circular fusion module as w/o MCF, removing triplets-prompt modality contrastive pre-training module as w/o PCL, and removing the aggregate relation context module is denoted as w/o AggRC. The results shown in Tables 4 and 5 demonstrate a marked decrease in performance with the exclusion of each module—underscoring their joint efficacy.

**Table 6: Evaluation results on YAGO15K dataset using various combinations of modalities**

| Model | YAGO15K | | | |
|---|---|---|---|---|
| | MRR | H@1 | Hit@3 | H@10 |
| S | 0.3594 | 0.2931 | 0.3925 | 0.4874 |
| S+V | 0.3819 | 0.3104 | 0.4240 | 0.5092 |
| S+T | 0.3866 | 0.3169 | 0.4258 | 0.5098 |
| **S+V+T** | **0.4303** | **0.3688** | **0.4617** | **0.5449** |

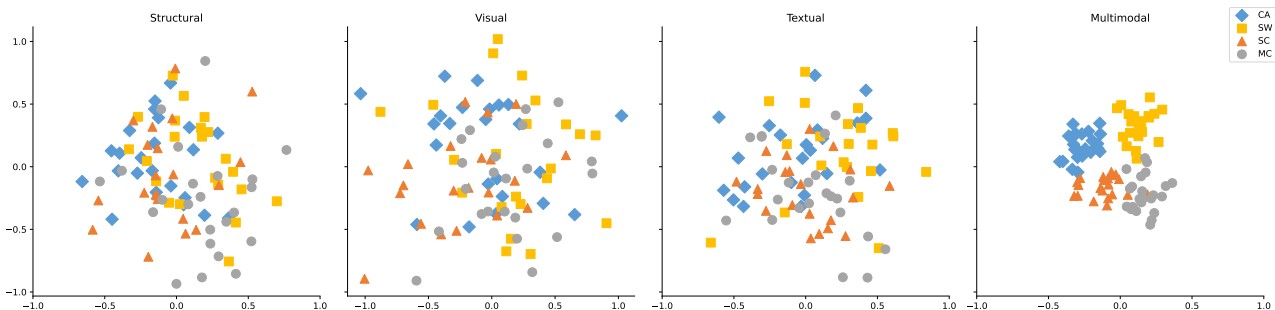

**Figure 4: Visualization of low-dimensional representations for football players under the context playsFor. Each colored node denotes a football player and the different colors denote four football clubs.**

**Table 7: Impact of Inter-Entity Multimodal Interactions on YAGO15K**

| Pre-Cross | Cross | YAGO15K | | | |
|:---:|:---:|:---:|:---:|:---:|:---:|
| | | MRR | H@1 | Hit@3 | H@10 |
| × | ×(Add) | 0.3967 | 0.3349 | 0.4282 | 0.5143 |
| × | ×(BP) | 0.4001 | 0.3361 | 0.4337 | 0.5154 |
| ✓ | ×(Add) | 0.3984 | 0.3338 | 0.4329 | 0.5133 |
| ✓ | ×(BP) | 0.4052 | 0.3468 | 0.4328 | 0.5200 |
| × | ✓ | 0.4194 | 0.3562 | 0.4506 | 0.5343 |
| ✓ | ✓ | **0.4303** | **0.3688** | **0.4617** | **0.5449** |

Specifically, the w/o PCL variant validates the necessity of pre-aligning modality semantics before modality fusion, which provides more harmonious modality features for subsequent interaction and fusion. Moreover, the w/o AggRC variant also illustrates the positive impact of considering relational context on link prediction tasks. It is particularly noteworthy that the removal of the Modality Circulant Fusion (MCF) module, which undertakes the main function of inter-entity modality interactions, has led to a significant decrease in performance. This demonstrates the effectiveness of employing simple yet effective multilinear transformations in modeling inter-entity modality interactions.

### 5.6 Modality Ablation Study

To validate the impact of modal information on link prediction accuracy enhancement, the modality ablation study is conducted, as shown in Table 6. This involved assessing the contributions of various combinations of modal embeddings, including structural information (S), visual information (V), and textual information (T), and the results clearly indicate that reliance on a single modality results in the least effective performance, while the incorporation of comprehensive multimodal information significantly improves outcomes. This emphasizes our model's proficiency in adeptly capturing multimodal information to elevate the performance of link prediction tasks.

### 5.7 Inter-Entity Modality Interactions Study

To verify the effectiveness of inter-entity modality interactions for link prediction, we carried out experiments as depicted in Table 7.

The experiments examined whether inter-entity modality interactions during the pre-training and training phases make a difference within MoCi framework.

- ×: Indicates scenarios without inter-entity modality interactions, focusing solely on intra-entity modality interactions.
- ×(Add): Signifies the additive fusion of modalities.
- ×(BP): Signifies the use of bilinear pooling used in IMF [34] for the fusion of modalities.
- (✓): Symbolizes PCL in Pre-Cross, MCF in Cross.

The experimental results indicate that, whether during pre-training or the training phase, results with inter-entity modality interactions consistently outperform those without such interactions. This not only validates the efficacy of our model's multimodal fusion approach but also underscores the significance of inter-entity modality interactions in enhancing link prediction accuracy.

### 5.8 Case Study

To highlight the effectiveness of our MoCi model, we employ $t-SNE$ for dimensionality reduction to visually represent the contextual embeddings of football players across different football clubs. Figure 4 shows that representations based solely on unimodal data tend to overlap, reflecting the inherent biases of this approach. In contrast, MoCi leverages circulant interactive multimodal fusion, effectively capturing the inter-entity modality semantics. This method demonstrates MoCi's advanced ability to discern complex inter-entity modality interactions, distinguishing football players' representations by their respective clubs more clearly.

### 6 CONCLUSION

Our paper introduces MoCi, a novel method that models multimodal contextual interactions of entities and achieves notable improved link prediction. MoCi addresses the deficiency of limited modal interactions within individual entities in most existing methods. Specifically, our proposed triplets-prompt modality contrastive pre-training strategy aligns modality semantics in advance. Moreover, explicit inter-entity modality interactions are effectively modeled by the proposed modality circular fusion method. MoCi represents the first endeavor to model multimodal contextual interactions from an inter-entity perspective. SOTA experimental results on seven datasets validate MoCi's efficacy in modeling inter-entity modality interactions for improving link prediction performance.

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
