# OpenReview forum: "Multimodal Contextual Interactions of Entities: A Modality Circular Fusion Approach for Link Prediction"
_acmmm.org/ACMMM/2024/Conference — MM2024 Poster_

### Official Review · Reviewer_yzzA · 2024-05-06

**Rating:** 4
**Confidence:** 2

**Summary:**

This paper introduces a novel approach called Modality Circular fusion (MoCi) for link prediction in knowledge graphs (KGs) by modeling inter-entity modality interactions. The key contributions include a triplets-prompt modality contrastive pre-training to align modality semantics, a modality circular fusion model for explicit inter-entity modality interactions, and relational context-aware predictions. The method is evaluated on seven datasets and achieves state-of-the-art performance.

**Strengths:**

1. MoCi presents a novel approach that focuses on inter-entity modality interactions, which is not extensively explored in existing methods.
Technical Approach: The use of multimodal contextual information and the proposed fusion strategy show technical sophistication.
2. The method is validated on seven datasets, demonstrating its effectiveness and superiority over existing approaches.

**Limitations:**

1. The complexity of the proposed approach, especially the modality circular fusion model, could potentially limit its computational efficiency, especially when applied to large-scale KGs or in real-time applications.
2. While the paper mentions explainability in the context of the circular fusion operation, a more thorough discussion or analysis of the interpretability of the learned representations and fusion process could be beneficial.
3. The paper does not discuss the data and resource requirements for implementing the proposed approach, including the size and nature of the KGs, the computational resources needed for training and inference, and the availability of pre-trained models for initialization.
4. It would be valuable to see an analysis of the robustness of the proposed approach to variations in hyperparameters, dataset characteristics, and modalities, as well as a sensitivity analysis to key design choices in MoCi.
5. The paper assumes that the multimodal data is of high quality and properly aligned, which might not always be the case in real-world scenarios. Addressing data preprocessing and cleaning could be a limitation.
6. The model is designed for three modalities (text, image, and structure), and it might be challenging to generalize it to other unseen modalities without additional adaptation or fine-tuning.

**Suitability:**

2

---

### Official Review · Reviewer_iQ6X · 2024-05-19

**Rating:** 3
**Confidence:** 3

**Summary:**

This paper focuses on the multimodal knowledge graph completion (MKGC) task, which aims to infer missing triplets based on the multimodal information of entities and relations from the knowledge graph. The paper emphasizes the importance of inter-entity interaction, i.e., interactions of different entities. To this end, this paper proposes a two-stage training method, MoCi, which first learns multimodal embeddings for entities and relations with contrastive pre-training, then uses modality circular fusion to model inter-entity modality interactions. The proposed MoCi is evaluated on seven MKGC datasets compared with various baselines.

**Strengths:**

1. Knowledge graph completion is an important task, and the utilization of multimodal content is helpful for real-world applications.
2. The proposed modality circular fusion is somewhat innovative.
3. The proposed method is compared with extensive baselines across seven datasets.

**Limitations:**

1. **The motivation shown in Figure 1 is somewhat confusing.** The authors mention that $t_{hoffman}$ is more semantically compatible with $v_{rain\ man}$ compared to  $v_{the\ lion\ king}$ (lines 115 to 133). However, this is not intuitive. It appears that $t_{hoffman}$ and $t_{rain\ man}$ are more compatible because the text contains keywords like "vulnerable" and "autistic savant." The authors should further clarify this point.
2. **The experimental results are not very convincing.** It appears that there are differences between some baselines and the results reported in the original papers, but the authors do not explain the specific reasons. For example, IMF has an MRR of 0.485 on DB15K [1], whereas the authors' result is 0.441. Furthermore, in Table 3, why are different encoders used for different datasets? Do other baseline models use the same encoders? The authors should clarify these points to avoid unfair comparisons.
3. **Lack of efficiency experiments and analysis.** The authors use a two-stage training method, which raises concerns about potentially long training costs. Additionally, there is no analysis of the time complexity for the proposed Modality Circular Fusion (MCF) module.
4. **Lack of analysis of parameter sensitivity.** How should the best hyperparameters be selected? This would help understand the stability of the proposed model.

More Questions:
1. In line 339, what does $r_r^p$ represent, and does each relation $r$ also have multimodal representations? The authors only explain entity features in Table 1 but do not introduce relation features.
2. Are the reported results from a single run or the average of multiple runs? If it is the latter, the authors should provide the standard deviation or p-value to indicate the significance.
3. Why are the baselines different across datasets? It seems that the comparisons with baselines are selective across different datasets.

Reference

[1] IMF: Interactive Multimodal Fusion Model for Link Prediction. WWW 2023.

**Suitability:**

3

---

### Official Review · Reviewer_vh3x · 2024-05-25

**Rating:** 2
**Confidence:** 3

**Summary:**

This paper proposes a method called MoCi for link prediction task, which aligns modality semantics through a triplets-prompt modality contrastive pre-training strategy and achieves modality interaction between entities through a modality circular fusion model. Extensive experiments on multiple datasets have demonstrated the effectiveness of this method.

**Strengths:**

1 This paper innovatively conducts inter-entity modality interactions, significantly enhancing the model's performance.
2 Extensive experiments and ablation studies validate the effectiveness of the proposed MoCi framework, showing a significant performance improvement over existing methods.

**Limitations:**

1 There are several terms in the paper that I find hard to understand, such as ” structural bias”,"semantic critical," "semantic compatibility," and "triplets-prompt." Is it appropriate to use expressions like "prompt"? The author should provide clear definitions for these terms.
2 How are valid triplets and invalid triplets determined? How is the difference in "semantic compatibility" reflected in the example of Figure 1? Without prior knowledge, it is impossible to determine the relationship between Dustin_Hoffman and Rain_Man and Lion_King.
3 The writing of the paper is confusing; 'bt' and the 'structural loss function' are mentioned in Section 4.2, but their definitions are provided in Section 4.4.

**Suitability:**

2

---

### Official Review · Reviewer_BEXm · 2024-05-25

**Rating:** 4
**Confidence:** 2

**Summary:**

This paper proposes a novel approach called ModalityCircular fusion (MoCi), which models inter-entity modality interactions by interweaving the multimodal contexts of entities.

Unlike most methods that directly fuse modalities, the authors design a triplets-prompt modality contrastive pre-training to align modality semantics beforehand. Additionally, they introduce a modality circular fusion model employing a simple yet efficient multilinear transformation strategy, allowing explicit inter-entity modality interactions and distinguishing it from methods that fuse modalities within individual entities.

**Strengths:**

1. Multimodal link prediction is an important foundational task.
2. The experimental results in this paper are very good. Although I reviewed some baseline papers (such as IMF) and found differences between the reported results and the original ones, I hope the authors can provide an explanation. However, even considering the original performance, the results in this paper are still impressive.

**Limitations:**

1. The provided code is unusable, with no executable code available. This undermines my confidence in the reproducibility of this paper.
2. The writing of this paper needs improvement, as the expression is not intuitive. For example, can it clearly restate the connection between the methods and the motivations of the paper?
3. The figures in this paper are not intuitive. There are many unclear parts in the motivation and method diagrams, especially in the motivation diagram, which is really hard to understand.
4. The reason I gave Borderline Accept is that the method presented in this paper is quite interesting. However, if other reviewers find significant issues in the method, I will lower my rating.

**Suitability:**

2

---

### Meta-Review · Area_Chair_KgxM · 2024-07-03

**Recommendation:** Accept (Poster)
**Confidence:** 4

**Metareview:**

The paper is mainly for multimodal link prediction, highlighting its importance and innovative approach, particularly through the proposed MoCi framework, which significantly enhances model performance by conducting inter-entity modality interactions. The extensive experiments and comparisons across seven datasets demonstrate the method's effectiveness. There were discrepancies in baseline results and a lack of explanations for these differences. Reviewers also noted issues with the figures, which were not intuitive and required clearer presentation. The complexity of the approach and potential computational inefficiencies were mentioned, along with the need for a more thorough discussion on interpretability, resource requirements, and robustness to variations in hyperparameters and dataset characteristics. Despite these limitations, the innovative nature of the method and its demonstrated performance improvements suggest that this paper should be accepted, provided the authors address the highlighted concerns.